# β-Lactams from the Ocean

**DOI:** 10.3390/md21020086

**Published:** 2023-01-25

**Authors:** Jed F. Fisher, Shahriar Mobashery

**Affiliations:** Department of Chemistry & Biochemistry, 354 McCourtney Hall, University of Note Dame, Notre Dame, IN 46656–5670, USA; jfisher1@nd.edu (J.F.F.); mobashery@nd.edu (S.M.)

**Keywords:** enzyme inhibitors, PBP, penicillin-binding protein, β-lactonase, salinosporamide, AHL, *N*-acylhomoserine lactone, quorum quenching

## Abstract

The title of this essay is as much a question as it is a statement. The discovery of the β-lactam antibiotics—including penicillins, cephalosporins, and carbapenems—as largely (if not exclusively) secondary metabolites of terrestrial fungi and bacteria, transformed modern medicine. The antibiotic β-lactams inactivate essential enzymes of bacterial cell-wall biosynthesis. Moreover, the ability of the β-lactams to function as enzyme inhibitors is of such great medical value, that inhibitors of the enzymes which degrade hydrolytically the β-lactams, the β-lactamases, have equal value. Given this privileged status for the β-lactam ring, it is therefore a disappointment that the exemplification of this ring in marine secondary metabolites is sparse. It may be that biologically active marine β-lactams are there, and simply have yet to be encountered. In this report, we posit a second explanation: that the value of the β-lactam to secure an ecological advantage in the marine environment might be compromised by its close structural similarity to the β-lactones of quorum sensing. The steric and reactivity similarities between the β-lactams and the β-lactones represent an outside-of-the-box opportunity for correlating new structures and new enzyme targets for the discovery of compelling biological activities.

## 1. Introduction

The discovery of antibacterial antibiotics revolutionized the practice of medicine [1,2]. Among the seminal structures isolated during the golden age of antibacterial discovery—the two decades following the realization in 1943 of the clinical efficacy of the penicillins—were bacitracin, polymyxins, vancomycin, and cephalosporins. These structures (including the penicillins) share three attributes: they are still used in modern medicine, their mechanism is the inhibition of the proper assembly of the bacterial cell envelope [3,4], and they originate as secondary metabolites of terrestrial organisms. This latter attribute has engendered (on multiple occasions) the question: might marine organisms also offer unique and transformative antibacterial structures? The potential value of marine organisms as a source of compelling, biologically active structures is no longer a premise, but a fact. Indeed, the impressive list of antitumor-antibiotic structures isolated from marine sources (didemnin, gephromycin, gliotoxin, grincamycin, ilimaquinone, lamellarin, largazole, lurbinectedin, meridianin, peloruside, phorbazole, plinabulin, staurosporine, trodusquemine, withanolide…) exemplifies both unique chemical structure and exceptional biological activity [5,6,7]. While the experience with another marine antitumor antibiotic (bryostatin) has underscored the extraordinary difficulty of translating exceptional in vitro activity to clinical relevance [8,9,10], it has also proven that marine-derived structures are “privileged” in that their structural diversification can uncover new clinical utilities [11,12]. Given that the future for antibacterial discovery is structured to counter infections caused by multi-drug-resistant bacteria, chemical entities that have privilege are prized. Recognition that the biosynthetic source of these marine structures is in many cases marine bacteria is complemented by a growing understanding of what bacteria to culture and how to genetically manipulate the bacteria. These questions regarding marine bacteria, as in the case of marine *Streptomyces* [13,14,15,16], may address the difficulty of understanding the ecological role of “antibiotic” secondary metabolites [17,18], so as to enable activation of their biosynthetic gene clusters. Practical solutions to this challenge were discussed recently with reference to lasonolide A, another as yet promising marine antitumor antibiotic [19,20].

The structural focus is the β-lactam ring—the four-membered cyclic amide sub-structure—of penicillin (Figure 1). The β-lactam ring confers the antibacterial biological activity, not just to the penicillins but to the other clinically important structural sub-classes which together comprise the diverse β-lactam antibiotic family. These sub-classes include cephalosporins, cephamycins, clavulanic acid, nocardicins, monobactams, and carbapenems [21]. All of these β-lactams are enzyme inhibitors. Most pathogenic bacteria contain a cell wall that is made biosynthetically from glycan strands, which have peptide stems on their alternate saccharides [22]. In the final stage of cell-wall biosynthesis, these stems are cross-linked so as to conjoin adjacent glycan strands. Cross-linking is the result of acyl transfer from one stem to the nucleophilic serine of the transpeptidase enzyme catalyst, followed by the transfer of the acyl moiety to an amine functional group of the adjacent glycan strand. Linkage of adjacent glycan strands across the surface of the bacterium creates an encasing cell-wall polymer, called the peptidoglycan. β-Lactams inactivate the transpeptidase enzymes of peptidoglycan biosynthesis [23]. Their β-lactam ring is recognized as structural mimetics of the endogenous stem peptide and acylates the active-site serine with the concomitant opening of the β-lactam ring. As the resulting acyl-enzyme is sterically incompetent for acyl-transfer [24], the catalytic activity of these transpeptidase enzymes—termed historically as “penicillin-binding proteins” (PBPs)—is lost. This loss of activity is bactericidal when bacteria are growing actively.

The marine antitumor antibiotic salinosporamide A (Figure 1) offers a structural counterpoint to the β-lactam sub-structure. Salinosporamide A (Marizomib^®^) is currently in late-stage clinical evaluation for brain cancer [25,26]. Its structural core is a β-lactone ring. In a mechanism similar to the β-lactams, its β-lactone is opened by the active-site threonine of the human 20S proteasome to give an acyl-enzyme species [27,28]. An ensuing second intramolecular ring-closing reaction displaces the chlorine as a chloride-leaving group and renders the acylation irreversible [29]. The biosynthetic complexity of the penicillins and the salinosporamides is comparable. Given that marine organisms biosynthesize the latter, do they also synthesize the former? 

In this essay, we address the following questions. Do marine organisms biosynthesize notable inhibitors of bacterial cell-wall biosynthesis? Do marine organisms biosynthesize classical antibacterial β-lactams? Do marine organisms biosynthesize (other) β-lactams? Is there a relationship between the β-lactam functional group and the β-lactone functional group, and the observation from the marine environment that marine bacteria have numerous β-lactam-hydrolyzing enzymes?

## 2. Do Marine Organisms Biosynthesize Exceptional Inhibitors of Bacterial Cell-Wall Biosynthesis?

At present, no marine structure having both the efficacy and safety to justify its clinical use has been identified. None of the structures shown in Figure 2 and Figure 3 meet this standard. All of the exploratory antibacterial antibiotics in current clinical trials have terrestrial origin [30,31,32]. However, the distinction between the marine and terrestrial origins of natural products must be done cautiously. While there are numerous examples of structurally and biologically unprecedented marine secondary metabolites (for example, the list of marine-derived antitumor antibiotics presented above), the two realms have substantial structural overlap [33]. Moreover, as is seen with respect to the antimicrobial evaluation of terrestrial secondary metabolites, the antimicrobial evaluation of marine secondary metabolites likewise identifies innumerable structures possessing modest to good antibacterial activity [34]. Inventories of these structures are published regularly [15,35,36,37,38,39,40,41,42,43,44,45,46,47,48]. Within these inventories are found structures that are truly distinctive, often in the multiple aspects of their biosynthesis, ring construction, and biological activity. 

The structures in Figure 2 exemplify distinctive marine-derived antibacterials. The abyssomicin family of macrolactones (more than thirty spirotetronate structures, exemplified here by abyssomicin C) is encountered as secondary metabolites of marine *Streptomyces* [49,50,51,52,53,54]. The abyssomicins have attracted extensive synthetic (notably, evaluation of the property of many members of this family, including abyssomicin C, to show room-temperature atropisomers) and biosynthetic studies. Abyssomicin C has potent Gram-positive (*Staphylococcus aureus*) antibacterial and antimycobacterial (*Mycobacterium tuberculosis*) activities (both, MIC 5 mg·L^−1^). In addition, the abyssomicins show human cell-line cytotoxicity. The enzyme target of the abyssomicins is 4-amino-4-deoxychorismate synthase (PabB), the enzyme catalyst for the synthesis of 4-aminobenzoic acid from chorismite [55]. Loss of PabB activity blocks folate coenzyme biosynthesis [50,53]. PabB inactivation by abyssomicin is initiated by the conjugate addition of the thiol of its catalytic cysteine to the enone of abyssomycin to give the enolate [55]. Intramolecular trapping of this enolate by the butenolide sub-structure of abyssomycin gives the stable, final covalent structure of inactivated PabB [56]. This mechanistic sequence is conceptually similar to the salinosporamides against their different enzyme target. 

Anthracimycin is a macrolide isolated from marine *Streptomyces* (but now also a terrestrial secondary metabolite) [57,58]. Anthracimycin also has engendered chemical interest. Its biosynthesis (and that of a cognate secondary metabolite, chlorotonil) is visualized as involving a spontaneous (while during PKS assembly) intramolecular [4 + 2] cycloaddition [59,60,61,62]. Anthracimycin shows in vitro MIC values of ≤0.25 mg·L^−1^ against all strains of *S. aureus* [63]. Notwithstanding a fundamental difference with respect to stereochemistry (Figure 3, compare the ring stereochemistry of anthramycin to that 2b-Epo), comparable antibacterial as well as antimalarial activities are found for chlorotonil derivatives. Culture of the chlorotonil-producing myxobacterium *Sorangium cellulosum* on a >150 L scale secured multi-gram quantities of chlorotonil. Its chemical stabilization (by reductive mono-dechlorination) followed by bis-epoxidation (to improve the water solubility to 16 µM) gave structure 2b-Epo, retaining essentially the full breadth of biological activities [64]. Structure 2b-Epo has in vitro MIC values of <0.05 mg·L^−1^ against several Gram-positive bacteria (including an MIC_90_ of 0.1 mg·L^−1^ against methicillin-resistant *S. aureus*), oral safety in mice at 50 mg·kg^−1^ attaining serum concentrations above the MIC for the activity for 24 h, and efficacy in the *S. aureus* mouse neutropenic thigh infection assay at i.v. 2 × 5 mg·kg^−1^ dosing [64]. Its mechanism is not yet known. Equisetin exemplifies a structure with a breadth of biological activities, among which is the inhibition of bacterial acetyl-CoA carboxylase resulting in failed fatty acid biosynthesis [65]. Its own biosynthesis involves an enzyme-catalyzed ring-forming Diels–Alder cycloaddition [66,67]. Equisitin has potent Gram-positive antibacterial activity [68,69] and shows pronounced synergy with polymyxins against pathogenic Gram-negative bacteria [68,70]. Marine bacteria have significant potential as producing organisms of the tripyrrole prodiginines [71]. The prodiginines (represented by prodigiosin) demonstrate a breadth of biological activities, also including pronounced synergy with polymyxins against pathogenic Gram-negative bacteria [72,73,74,75].

The structures in Figure 3 exemplify a second set of distinctive marine-sponge-derived antibiotics. This figure shows structures that act to inhibit different targets within bacterial cell-wall biosynthesis. Dibromoageliferin represents one structure within the large and diverse class of bromopyrrole-imidazolamine structures (including also the sceptrins, the oroidins, and the nagelamides) [34]. As a class, these structures show potent Gram-positive and Gram-negative antibacterial activity [76,77,78]. They interfere with multiple stages of cell-envelope biosynthesis, including membrane integrity, assembly of the cytoskeleton, and peptidoglycan integrity [79,80]. The alga-derived chrysophaetins target the FtsZ protein of the Gram-positive cytoskeleton [81,82,83]. Taromycin A is recognized immediately as a marine-derived cognate structure of the Gram-positive antibiotic, daptomycin [84,85]. Daptomycin interferes with bacterial peptidoglycan biosynthesis by complexation with its biosynthetic intermediates [86,87]. Daptomycin is used increasingly in the clinic against serious Gram-positive bacterial infections. The marine origin of dibromoageliferin, chrysophaetin A, and taromycin A is attested to by their halo substituents.

Labdanes are secondary metabolites of both terrestrial and marine *Streptomyces* [88,89,90]. Although weakly antibacterial against methicillin-resistant *S. aureus* (MRSA, MIC 32–64 mg·L^−1^), as an inhibitor of the FEM enzymes unique to *S. aureus* [91], cyslabdan synergy reduces the in vitro MIC of carbapenems against MRSA by up to 1,000-fold (in the presence of 10 mg·L^−1^ cyslabdan, the MIC of imipenem is reduced from 16 mg·L^−1^ to 0.015 mg·L^−1^). The potentiation of the MICs of β-lactams was much less (32-fold for penicillins, 4–32-fold for cephalosporins) [92], implicating a high correlation between the specific PBP inactivated by the β-lactam and synergistic inhibition of the FEM enzymes by cyslabdan. β-Lactam synergy was observed with a marine-derived cyslabdan (isolated from a different marine *Streptomyces*) across a panel of Gram-positive and Gram-negative bacteria. The basis for the synergy was attributed (in part) to the inhibition of the β-lactam-hydrolyzing (β-lactamase) activity of the panel [93]. Remarkably, this interpretation has not received subsequent verification, possibly as a result of the separation between laboratories having access to the marine-derived structure, and laboratories with the ability to carry out a rigorous mechanistic study using enzymes from notable bacterial pathogens. Synthetic access to cyslabdan structures is, however, established [94]. Lipoxazolidinones are a class of marine-derived Gram-positive antibiotics [95,96]. They have several outstanding attributes: potent Gram-positive antibacterial activity (MRSA strains, MIC ≤2 mg·L^−1^), dual mechanisms of action (inhibition of both peptidoglycan and protein biosynthesis) with low frequency of resistance mutation, and accessibility to synthetic modification, with the possibility of expansion of their antibacterial activity to Gram-negative bacteria [97]. The relationship between their structure to their molecular mechanism is not known. Their oxazolidinone ring suggests the possibility of target acylation, as seen for the β-lactams and the salinosporamides (but not a mechanistic aspect of the better-known oxazolidinone Gram-positive antibiotics, exemplified by linezolid).

The two final structures in Figure 3 are marine-derived inhibitors of β-lactam antibiotic resistance enzymes of bacteria, the β-lactamases. Kalafungin (isolated from marine *Streptomyces*) is a weak (IC_50_ = 225 µM) inhibitor of Gram-positive β-lactamases [98]. Halisulfate-5 is a more potent inhibitor (*K*_i_ = 6 µM) of the clinically much more relevant AmpC β-lactamases of Gram-negative bacteria [99]. The crystal structure of the halisulfate-5·AmpC complex opens the opportunity for structure-based design. Recognition that marine sources produce inhibitors of these β-lactamases (as assessed in an in vitro assay) suggests two conclusions: that marine organisms might biosynthesize β-lactams, and that these β-lactamases are present as a resistance mechanism to these β-lactams. As we discuss, neither conclusion has decisive experimental support.

## 3. Do Marine Organisms Biosynthesize The Classical Antibacterial β-Lactams?

It is uncertain whether marine organisms biosynthesize classic β-lactam antibiotics. A momentous event in the history of the β-lactams was the isolation by Brotzu in 1945—from the Mediterranean Sea, near a sewage outfall located at Cagliari, Sardinia—of the *Cephalosporium acremonium* fungus, which biosynthesizes cephalosporin C. Notwithstanding the singular importance of his discovery, no antibacterial β-lactam has been isolated since from a marine source. The subsequent β-lactam sub-families—nocardicins, clavulanic acid, monobactams, and carbapenems—discovered over the course of ensuing decades are secondary metabolites of terrestrial bacteria. While numerous biologically active secondary metabolites are isolated from marine *Penicillium* fungi, none is a β-lactam [45,100]. A 2003 analysis of the genomic DNA of the marine fungus *Kallichroma tethys* identified two genes (*pcbAB* and *pcbC*) encoding proteins homologous to the penicillin biosynthetic enzymes of *Acremonium chrysogenum* [101]. While circumstantial evidence suggested that these genes were regulated and expressed, no antibiotic was identified in its culture. This observation has not been pursued. This absence of interest may reflect the remarkable accomplishment of the industrial-scale production of penicillins and cephalosporins from the antecedents of their original producing fungi. It would appear that there is no commercial need for a marine-derived producing organism of these antibiotics. If this explanation is correct, the result is unfortunate. Contrary to the surmise that the β-lactam represents a challenging functional group for biosynthesis, the biosynthetic pathways leading to the individual β-lactam sub-families are astonishingly diverse [102,103,104]. Nature has devised multiple pathways for the biosynthesis of β-lactam. Moreover, with the exception of nocardicins, each β-lactam sub-family has achieved a dramatic impact in the chemotherapy of bacterial infections. If a marine producer of a new sub-family of the antibiotic β-lactams is discovered, this discovery could be equally transformative.

## 4. Do Marine Organisms Biosynthesize β-lactams?

Marine organisms biosynthesize other β-lactams. However, the structural exemplification is sparse (Figure 4). Antibiotic X372A was isolated in 1975 from a marine *Streptomyces* bacterium [105]. Its Gram-positive and Gram-negative antibacterial activity is the result of an ATP-dependent inhibition of glutamine synthetase, and not from inhibition of bacterial cell-wall synthesis. The mechanism of this inhibition, as studied with the related β-lactam structure tabtoxinine-β-lactam (“tobacco wildfire toxin”) biosynthesized by terrestrial *Pseudomonas* bacteria [106,107], does not involve the opening of its β-lactam ring [108,109]. Chartelline B was isolated in 1987 (as one of several related structures) from the marine bryozoan *Chartella papyracea*. It is not described as yet as having a biological activity [110]. Monamphilectine A was isolated (as one of several related structures) in 2010 from a Caribbean *Hymeniacidon* sp. sponge [111]. The isocyanide-containing monamphilectines have potent in vitro activity against the malaria-causing *Plasmodium falciparum* parasite [112]. The molecular target is not known.

## 5. Does the Marine Environment Contain β-lactam-degrading Enzymes?

A prominent role for marine-biosynthesized, β-lactam-containing enzyme inhibitors is not yet supported. To this date, three β-lactam-containing natural products are identified as biosynthesized by marine organisms. None is an inhibitor of bacterial cell-wall biosynthesis. To date, marine organisms biosynthesize several compounds that are adjuvants of the antibiotic activity of the terrestrial β-lactams (and possibly, other antibiotics), and several structures that inhibit in vitro the resistance enzymes, which hydrolytically degrade terrestrial-derived β-lactam antibiotics. These few examples could be interpreted to signify that the β-lactam functional group lacks significance within marine biology. Such an interpretation might follow the expectation that within the marine environment, the enzyme catalysts capable of the hydrolytic degradation of the terrestrial β-lactam antibiotics are not necessary and thus are uncommon. The evidence that this conclusion is incorrect is overwhelming. The oceans teem with such enzymes.

Two reasons support the prevalence of β-lactam-degrading enzymes in marine environments. The first reason is the copious and undisciplined use of antibiotics in human and animal medicine. The result is a profound “anthropogenic pollution” of the marine environment [113]. Antibiotic-resistance genes in the marine environment are now pervasive [114,115,116,117,118,119]. While the enzymes of antibiotic resistance—both terrestrial and marine [120,121]—are ancient, anthropogenic pollution has catalyzed their distribution. In anthropogenic-polluted marine environments, antibiotic-resistance enzymes are necessary for the survival of indigenous bacteria.

The basis for a second reason begins with the reminder that a principal basis for the resistance of bacteria to β-lactam antibiotics is the production of hydrolytic enzymes. These enzymes divide between those using an active-site serine nucleophile (Class A, C, and D) and those using zinc-ion catalysis (Class B, the metallo-β-lactamases). The serine β-lactamases are related evolutionarily to the penicillin-binding proteins of cell-wall biosynthesis, again with an ancient evolutionary separation of the β-lactamases from the penicillin-binding proteins [122,123]. The penicillin-binding protein motif is found additionally in esterase enzymes [124,125] and in biosynthetic transpeptidases [126,127]. Moreover, some of these esterases hydrolyze β-lactam antibiotics [128,129]. This mechanistic promiscuity underscores a fundamental difficulty in using in vitro enzymatic activity as a basis for enzyme nomenclature or presupposing a catalytic purpose for the enzyme. The magnitude of this difficulty is further emphasized by yet another aspect of bacterial ecology, that of quorum sensing [130]. Indeed, the recognition of quorum sensing as a phenomenon was made first with respect to the initiation of bioluminescence by marine *Vibrio* bacteria [131,132]. Quorum sensing offers a possible second reason for the extensive presence of β-lactam-hydrolyzing enzymes in the marine environment.

The context to understand this possible relationship begins with the chemical structures of the two ring systems introduced already: the β-lactams and the β-lactones. These rings have commonalities beyond ring size. β-Lactone structures are successful affinity probes of the penicillin-binding proteins (PBPs) [133,134,135]. The product of Class D β-lactamase hydrolysis of carbapenems is a β-lactone [136,137]. To these two rings may be added the isoxazolidin-3-one ring. This ring is encountered in lactivicin, which is also an inhibitor of the penicillin-binding proteins and the serine β-lactamases [138,139]. Each of these three rings (Panel A in Figure 5) is imbued with particular reactivity for the acylation of a nucleophile [139,140,141,142], and this reactivity is used to inhibit an array of enzymes [143]. The fourth ring of Panel A in Figure 5—dihydrofuran-2(3*H*)-one—while having lower intrinsic reactivity due to the absence of ring strain—is the ring system of the *N*-acylhomoserine lactone (AHL) class of quorum-sensing elicitors [130,131,132,144]. AHL quorum sensing does not use enzyme acylation. As quorum sensing correlates frequently with virulence, the identification of inhibitors of quorum sensing is an important objective [145,146]. While the favored *N*-acyl moiety is bacterial species-dependent (the AHL depicted in Panel B in Figure 5 is the AHL used to initiate marine *Vibrio* bioluminescence), the homoserine lactone ring is common to the AHL class of structures. One common transformation to abolish AHL signaling (a process termed quorum quenching) is the hydrolysis of the AHL amide with the release of the fatty acid. A second common transformation to achieve quorum quenching is hydrolytic ring-opening of the lactone. This reaction is catalyzed by the AHL lactonases (as further shown in Panel B in Figure 5). AHL lactonases are ubiquitous in marine bacteria [147]. The similarity between the hydrolytic reaction catalyzed by the AHL lactonases, and the hydrolytic reaction catalyzed by the β-lactamases, is evident (Panel B in Figure 5). This similarity challenges the classification of the serine-dependent enzymes and the metal-dependent enzymes found in marine bacteria. When such enzymes are purified and annotated as β-lactamases (and often as having an ancient heritage) [148,149,150,151,152], is the basis for their heritage that of β-lactam resistance or that of quorum quenching [153]? Selleck et al. suggest credibly that lactonase/lactamase promiscuity may offer an evolutionary advantage [153].

One example merits further comment. A class A β-lactamase produced by a bacterium at 1050 m below the surface of the Pacific Ocean turns over penicillins at or near the diffusion limit. This level of catalytic competence cannot be adventitious, implying a directed evolution for the purpose [149]. In light of the fact that the gene sequence for this enzyme shares the same GC content as the other genes within the genome of this bacterium, the anthropogenic origin in this case was ruled out. This enzyme was argued as the first bona fide β-lactam-resistance enzyme from a marine source. The antibiotic-resistance enzyme could be perceived as a countermeasure against organisms that produce β-lactam antibiotics within the niche in the depths of the ocean.

The intrinsic reactivity of β-lactones has been exploited to identify other antibacterial enzyme targets [141,154]. Several synthetic β-lactones were examined for their quorum-sensing activity. One β-lactone cognate structure (Figure 6, 1421598-00-6) lacked activity as an autoinducer of *Pseudomonas aeruginosa* quorum sensing [155]. A second β-lactone (Figure 6, 2021255-49-0) inhibits *Vibrio* quorum sensing, but also as a result of enzyme inactivation within the fatty-acid biosynthetic pathway and not within quorum pathways [156]. Antibacterial β-lactones act by inactivation of the ClpP protease of Gram-positive bacteria [157,158,159], and have been used to interrogate the biological mechanism of AHL-structure type eukaryotic human immune modulation [155,160,161]. Lastly, sub-structure searching of the AHL β-lactone structure returns as close structures salinosporamide (Figure 1) and obafluorin (Figure 6, a secondary metabolite of *Pseudomonas fluorescens*) [162]. The Gram-positive and Gram-negative antibacterial activity of obafluorin was identified recently as the result of the inhibition of bacterial threonyl-tRNA synthetase [163,164]. The structural similarity among obafluorin, salinosporamide, the AHL autoinducers, and obafluorin was noted previously [165]. To our knowledge, neither salinosporamide nor obafluorin has been examined for enzyme inhibition within the quorum or cell-wall biosynthesizing pathways.

In this essay, we discuss evidence to support the possibility that promiscuity with respect to the substrate—β-lactone or β-lactam—for marine β-lactonases/β-lactamases may explain (in part) a diminished advantage for a marine fungus or bacterium to produce a β-lactam antibiotic. A similarity in the chemical reactivity for the β-lactone and β-lactam may suggest value to examining marine β-lactones as new inhibitors of cell-wall biosynthesis. Both suggestions fit within “outside-the-box” approaches [166] to ensure a future for antibacterial discovery [167]. Further evidence is needed to support this possibility. Such support could inaugurate a new research field in the area of marine antimicrobial drugs.

## Figures and Tables

**Figure 1 marinedrugs-21-00086-f001:**
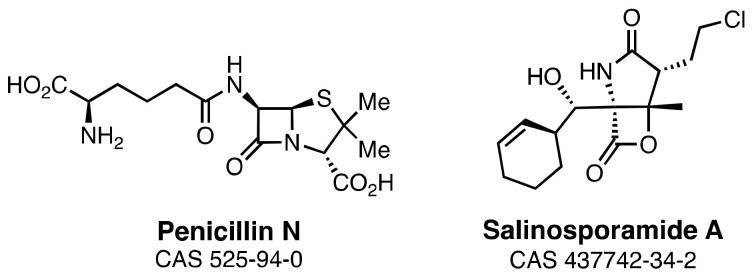
Structures of the terrestrial-derived β-lactam antibacterial antibiotic penicillin N and the marine-derived β-lactone antitumor antibiotic salinosporamide A.

**Figure 2 marinedrugs-21-00086-f002:**
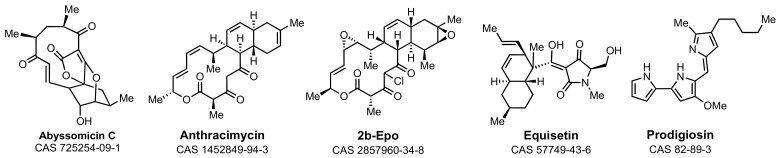
Distinctive marine-derived antibiotic structures with mechanisms other than interference with the enzymes of bacterial cell-wall biosynthesis.

**Figure 3 marinedrugs-21-00086-f003:**
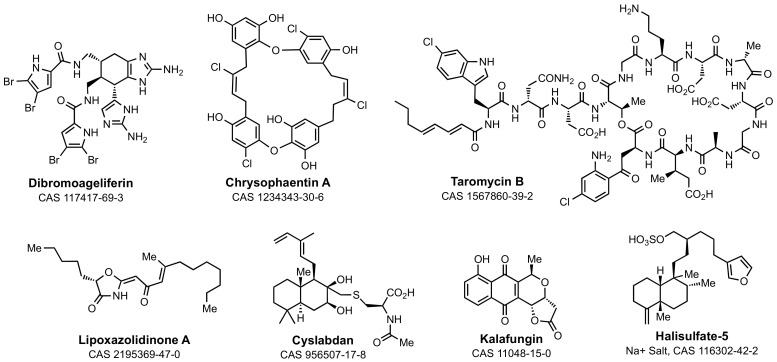
Distinctive marine-derived antibiotic structures with mechanisms involving inhibition of bacterial proteins. Dibromoageliferin, chrysophaentin A, taromycin B, lipoxazolidinone A, and cyslabdan inhibit key proteins involved in the synthesis of the bacterial cell envelope. Kalafungin and halisulfate-5 are inhibitors of the β-lactam antibiotic-resistance enzymes of bacteria, the β-lactamases.

**Figure 4 marinedrugs-21-00086-f004:**
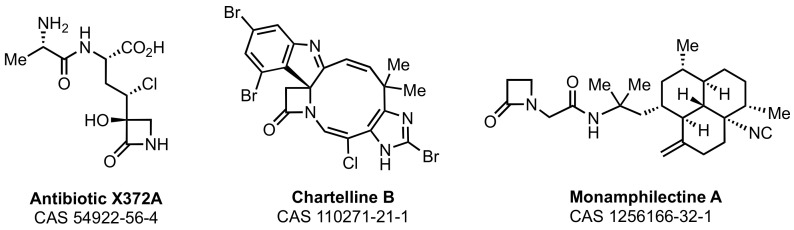
Structures of the marine-derived β-lactams.

**Figure 5 marinedrugs-21-00086-f005:**
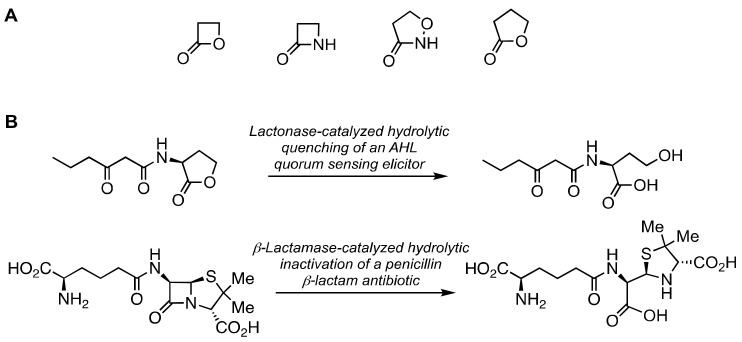
Panel (**A**), small rings activated for acyl-transfer. Panel (**B**), a comparison of the hydrolytic reactions catalyzed by the AHL quorum-quenching β-lactonases (upper reaction) and the β-lactamases (lower reaction).

**Figure 6 marinedrugs-21-00086-f006:**
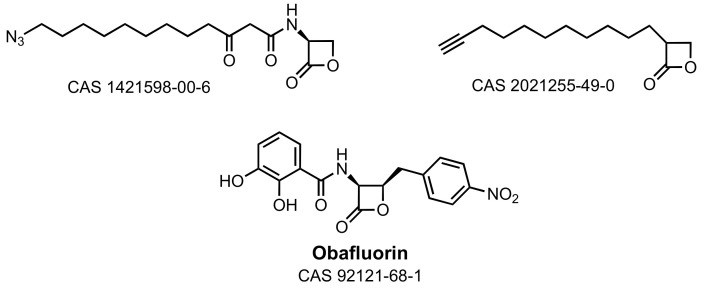
β-Lactone inhibitors of (top) bacterial fatty-acid biosynthesis and bottom (obafluorin) of threonyl-tRNA synthetase.

## Data Availability

Not applicable.

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
