# Peer review of "β-Lactams from the Ocean"

_marinedrugs, 2023, doi:10.3390/md21020086_

Round 1

Reviewer 1 Report

 The article by Jed F. Fisher and Shahriar Mobashery presents a review of the evidence suggesting that β-lactam antibiotics, including the penicillins, cephalosporins, and carbapenems are seldom, if at all, produce by marine microorganisms. They also review the antibiotics produced by marine microorganisms affecting cell-wall integrity, as well as bioactive compounds of marine origin which have a similar structure to β-lactam antibiotics, as well as the evidence suggesting that β-lactam-like compounds are frequent in marine environments since enzymes that are related to β-lactamases are frequent among marine organisms. They conclude that since quorum-sensing response dependent on acyl-homoserine lactones is prevalent among marine bacteria it is very much likely that the similarity of the β-lactam compounds with these auto inducer molecules, makes the coexistence of both type of molecules inoperative as they might interfere with each other metabolism. Furthermore, these authors propose that antibiotics interfering with cell wall synthesis might have a β-lactone structure in the marine environment.

I consider that the conclusions of this article are interesting, but there is still not sufficient information to sustain them. Therefore, I consider that it should be modified to clearly present their claims as hypothetical. In addition, I consider that the way the article is written is too colloquial for a scientific paper. 

I include some suggested changes, but the authors should revise the whole manuscript to address the points that I have raised.

Page 2

1.     Line 53. Change "of this essay" to "in which this article is centered".

2.     Line 59. Change "surround themselves with a protective cell wall. This cell wall " to "contain a cell wall which".

3.     Line 68. Insert "the" after "with" and change "comitant" to "concomitant".

4.     Line 71. insert "when bacteria are actively growing" after "bactericidal".

5.     Line 75. Change "While the β-lactam ring is the structural center of this essay, one final" to "As mentioned, we will focus in this essay in the β-lactam ring structure, so it is worth mentioning the"

6.     Line 76. Insert "which" before "provides" and "of this structure" after "counterpart".

7.     Line 84. Star a new paragraph after "former?" which says "In this article we...

8.     Line 88. Change "from the marine environment that its bacteria" to "that marine bacteria"

Page 3

9.     In the section starting on page 3 I suggest that you reduce to a minimum the description of compounds included in Chart 2, since they are not related to the main topic of the revision.

10.  Line 92. I suggest that the following sentences:

"Do marine organisms biosynthesize exceptional inhibitors of bacterial cell-wall biosynthesis? 

No. This answer is qualified by the word “exceptional”. Here, “exceptional” is de-92 fined as structures whose safety and efficacy justify human clinical evaluation." to:

"No cell-wall biosynthesis inhibitor produced by a marine organism that can be used in clinical settings has been identified.

At present, no structure that inhibits cell wall biosynthesis produced by marine organisms, whose safety and efficacy justify its evaluation in clinical trials has been identified."

11.  Line 113. Change the sentence "The abyssomicins are also show human cell-line cytotoxicity." to:

"However, the abyssomicins also show human cell-line cytotoxicity".

12.  Line 114. Change "They use their enone substructure as an" to "The enone substructure of abyssomicyn acts as an"

13.  Line 115. The sentence "Conjugate addition of a thiol (an essential cysteine residue of an enzyme) to this enone is followed by enolate trapping (by its butanolide) to give a final stable covalent structure " is not clear, rewrite it.

Make clear to which enzyme you are referring when you say an essential cysteine of "an enzyme".

14.  Line 120. Change "Its" to "The"

15.  Line 122. Start a different paragraph starting with "Anthracimycin is a macrolide"

Page 5.

16.  Lines 206-207. Change "Perhaps. This ambiguous answer is explained." to "It is not clear whether marine organisms synthesize classic antibiotic considering the following arguments:"

17.  Line 218. Delete "biosynthetic" between "no" and "antibiotic"

Page 6. 

18.     Line 228. Delete "(when?)" between "If" and "a".

19.     Line 230. Delete parenthesis from “other”.

20.     Line 231. Change "Yes" to "Marine organisms have been shown to produce other type of β-lactams."

4.     Lines 257. Delete "The answer to the question: Yes! 

Two reasons address this answer."

And start the second paragraph with: "There are two reasons that support that β-lactam degrading enzymes are prevalent in marine environments."

Page 8.

5.      Line 335. Start the last paragraph with:

"In this article we provide evidence to support the possibility that the promiscuity..."

6.     Line 340. End the last paragraph with the sentence:

"Further experimental evidence is needed to support these possibilities, and if our suggestions are further supported, a new research field in the area of marine antimicrobial drugs"

Author Response

Please see the attachments. The docx is a summary of the manuscript changes. The pdf highlights (yellow) the important text changes to the manuscript, in reply to the suggestions of the reviewer.

Reviewer 2 Report

This manuscript poses the question whether β-Lactams exist as marine secondary metabolites possessing exceptional inhibitory activity of cell-wall biosynthesis. Notwithstanding the given answer (NO), the authors suggest, based on the similarity of the chemical reactivity for the β-Lactone and β-Lactam rings, the value to examining marine β-Lactones as new inhibitors of cell-wall biosynthesis. Although very few typos were detected, for example α-lactone instead of a-lactone on line 78 of page 2 and  the redundant are on line157 of page 4, a minor spell-check may be required to render the present manuscript most suitable for publication. 

Author Response

Please see the attachment (same docx as for Reviewer 1, reply to Reviewer 2 is at the end).

Round 2

Reviewer 1 Report

The manuscript was modified according with my suggestions and I consider that it can reaccepted for its publication in its present form.